

# Simulation of crop yield using the global hydrological model H08 (crp.v1)

Zhipin Ai[1], Naota Hanasaki[1]

[1]Center for Climate Change Adaptation, National Institute for Environmental Studies, 16-2, Onogawa, Tsukuba 305-8506, Japan

*Correspondence to*: Zhipin Ai (ai.zhipin@nies.go.jp)

**Abstract.** Food and water are essential for life. A better understanding of the food–water nexus requires the development of an integrated model that can simultaneously simulate food production and the requirements and availability of water resources. H08 is a global hydrological model that considers human water use and management (e.g., reservoir operation and crop irrigation). Although a crop growth sub-model has been included in H08 to estimate the global crop-specific calendar, its performance as a yield simulator is poor, mainly because a globally uniform parameter set was used for each crop type. Here, through country-wise parameter calibration and algorithm improvement, we enhanced H08 to simulate the yields of four major staple crops: maize, wheat, rice, and soybean. The simulated crop yield was compared with the Food and Agriculture Organization (FAO) national yield statistics and the global data set of historical yield for major crops (GDHY) gridded yield estimates with respect to mean bias (across nations) and time series correlation (for individual nations). The improved simulations showed good consistency with FAO national yield. The mean biases of the major producer countries were considerably reduced to -4%, 3%, -1%, and 1% for maize, wheat, rice, and soybean, respectively. The corresponding coefficients of determination ($R^2$) of the simulated and FAO statistical yield increased from 0.01 to 0.98, 0.21 to 0.99, 0.06 to 0.99, and 0.14 to 0.97 for maize, wheat, rice, and soybean, respectively; the corresponding root mean square error (RMSE) decreased from 7.1 to 1.1, 2.2 to 0.6, 2.7 to 0.5, 2.3 to 0.3 t/ha. Comparison with the reported performances of other mainstream global crop models revealed that our improved simulations have comparable ability to capture the temporal yield variability. The grid-level analysis showed that the improved simulations had similar capacity to GDHY yield, in terms of reproducing the temporal variation over a wide area, although substantial differences were observed in other places. Using the improved model, we confirmed that an earlier study on quantifying the contributions of irrigation on global food production can be reasonably reproduced. Overall, our improvements enabled H08 to estimate crop production and hydrology in a single framework, which will be beneficial for global food–water–land–energy nexus studies.





## 1 Introduction

Food security has become an important global challenge because of the growing population and increasing competition for crop usage (Ray et al., 2022). A key factor in food security is crop production, which is largely affected by irrigation water availability, particularly in regions with insufficient precipitation (Chiarelli et al., 2022). Currently, for example, approximately 40% of global crop production relies on irrigation (Perrone et al., 2020). The use of water for this irrigation causes approximately 65% of global total water withdrawal and 90% of global water consumption (Shiklomanov, 2000; Döll and Siebert, 2002). These high rates of withdrawal and consumption have negative consequences for both surface water and groundwater systems, such as river fragmentation and groundwater table declines (McDermid et al., 2021; Perrone et al., 2020). To minimize such negative consequences, there is an increasing impetus toward sustainable water use (McDermid et al., 2021; Perrone et al., 2020; Rosa et al., 2018, 2020; Okada et al., 2018; Ai et al., 2021). To more fully address the complex interactions between crop production and sustainable water management, accurate representations of crop growth and water cycle with human activities should be placed within a consistent model framework during the development of an integrated model.

Many models have successfully incorporated the crop growth process and can simulate the global crop yield. These include LPJmL (Bondeau et al., 2007; Fader et al., 2010), GEPIC (Liu et al., 2007), PEGASUS (Deryng et al., 2011), CLM-Crop (Drewniak et al., 2013), PRYSBI2 (Sakurai et al., 2014), pAPSIM (Elliott et al., 2014), pDSSAT (Elliott et al., 2014), CROVER (Okada et al., 2015), ORCHIDEE-crop (Wu et al., 2016), PEPIC (Liu et al., 2016), and MATCRO (Masutomi et al., 2016). However, only a few of these models, such as LPJmL and CROVER, have globally implemented schemes for irrigation constrained by spatiotemporal detailed water availability (i.e., explicit consideration of river routing and water withdrawal). The lack of inclusion of such schemes severely limits the ability of these models to be used in comprehensive investigations of global food–water tradeoffs, particularly in terms of specifying the sources of water withdrawal used for crop irrigation.

In this study, we developed a new crop–water global model based on the H08 global hydrological model (Hanasaki et al. 2008a; 2018). Although H08 has detailed functions for specifying water sources and estimating crop specific yield based on the formulations of the SWAT model (Neitsch et al., 2002), its performance as a crop yield simulator has been poor in comparison with the FAO yield statistics and other gridded yield data sets. This poor performance is mainly because of the adoption of the global uniform parameters related to crop growth. These default parameters are acquired from the SWIM model, a variant of the SWAT model (Arnold et al., 1994), which is mainly for use in Europe and temperate climate zones (Krysanova et al., 2000). This leads to overestimation or underestimation when it is used in other regions with different crop management practices and climatic conditions. Additionally, the effects of $CO_2$ fertilization (Stockle et al., 1992) and changes in vapor pressure deficit (Stockle and Kiniry, 1990) on crop yield have not yet been considered. These two factors are particularly important in analyses the impacts of climate change on crop yield (Jägermeyr et al., 2021; Yuan et al., 2019).

Despite multiple attempts to optimize the parameters involved, global crop yield simulation remains challenging. For example, Fader et al. (2010) proposed the concept of management intensity, which represents the degree and frequency of field agronomy management (e.g., fertilizer, technology, and weed control). They adopted this concept in a global vegetation model, LPJml, by adjusting a key parameter of maximum leaf area index at the country level, which exhibited good agreement between the calibrated yield and FAO yield statistics. This adjustment enabled LPJml to be used in investigations of the crop–water relations by estimating crop water productivity and virtual water content (Fader et al., 2010). Deryng et al. (2011) calibrated the light use efficiency coefficient based on spatially explicit crop yield data reported by Monfreda et al. (2008). Iizumi et al. (2009) developed a large-scale crop model for paddy rice in Japan, known as the PRYSBI model, whereby multiple parameters were calibrated via the Markov Chain Monte Carlo technique at subnational level. The results showed that the Markov Chain



Monte Carlo method is a powerful approach for optimizing multiple parameters in a nonlinear and complex model. Sakurai et al. (2014) used a similar method globally and estimated eight parameters based on Free-Air Carbon Dioxide Enrichment (FACE) data with hundreds of thousands of calculation steps in the Markov Chain Monte Carlo process. Each of the above methods has its own advantages and disadvantages. For example, the method of Fader et al. (2010) was based on FAO national yield statistics, whereas the methods in the other three studies require spatial explicit yield data. Additionally, Fader et al. (2010) and Deryng et al. (2011) mainly focused on a single parameter, whereas Iizumi et al. (2009) and Sakurai et al. (2014) addressed with multiple parameters.

To enhance the capacity of H08 to simulate the yields of four major staple crops (i.e., maize, wheat, rice, and soybean), we first added two new functions to the H08 crop sub-model by considering the effects of $CO_2$ fertilization and vapor pressure deficit change on crop yield. Then, we adopted the method of Fader et al. (2010) for parameter calibration because of its robust performance, minimal computation cost, simplicity of implementation, and easily accessible and generally reliable input yield data when implemented in a global scale process-based crop growth model. Next, we evaluated model performance with respect to mean bias, time series variation, and time series correlation in accordance with the general framework proposed by Muller et al. (2017), using FAO statistical national data and recently published gird-level data. We sought to determine whether the improved H08 model could reproduce the mean historical yield at national scale; to determine whether the model could also capture interannual variation in historical yield times series; and to compare spatial time series correlations with other spatial explicit data. Finally, we investigated the contributions of irrigation to the global production of maize, wheat, rice, and soybean using the improved model as a case study for its application.

## 2 Materials and methods

### 2.1 H08 overview

H08 is a global hydrological model that includes natural and anthropogenic hydrological processes at a spatial resolution of 0.5° and a temporal resolution of 1 day. It was developed with six sub-models: land surface hydrology, river routing, crop growth, reservoir operation, environmental flow requirements, and anthropogenic water withdrawal (Hanasaki et al., 2008a). It has been updated with several new schemes including groundwater recharge and abstraction, aqueduct water transfer, local reservoirs, seawater desalination, and return flow and delivery loss (Hanasaki et al., 2018). With these newly added functions, H08 is one of the most detailed global hydrological models available for the estimation of sector-wise and water source-wise water withdrawal and availability. In the agriculture sector, H08 can estimate irrigation water demand and supply on a daily and grid-cell basis with several unique features. First, it can estimate the irrigation water withdrawal from both renewable and non-renewable groundwater sources. Second, it considers the effects of irrigation water withdrawal in the upper stream. Third, it includes the influence of reservoir operation on irrigation water availability. H08 was fully described in multiple previous studies (Hanasaki et al., 2008a, 2008b, 2018).

### 2.2 Crop sub-model

The crop growth sub-model accumulates plant biomass at a daily interval until physiological maturity; it also simulates phenological development. The daily increase in potential biomass ($\Delta B$) (kg ha$^{-1}$) is estimated based on radiation use efficiency and photosynthetic active radiation, using the method of Monteith et al. (1977) (see Eq. 1). Crop phenological development is based on daily heat unit accumulation theory, whereby physiological maturity is reached when the accumulated daily heat unit value is equal to the potential heat unit value. The harvest index is used to partition the total aboveground biomass with respect to grain yield. Regulating factors, including water and air temperature, are used to adjust the yield variation. Although the algorithm is based on SWAT and SWIM, and a detailed description was previously provided (Hanasaki et al., 2008a; Ai et al.,



2020), the main formulation is briefly described here because it is an important foundation for the forthcoming discussion on parameter optimization. In specific,

$$\Delta B = be * PAR * REGF \tag{1}$$

where $be$ is a crop-specific parameter of radiation use efficiency, $PAR$ is photosynthetically active radiation, and $REGF$ is the

crop regulating factor. $PAR$ is calculated using shortwave radiation ($Rs$) (W m$^{-2}$) and leaf area index ($LAI$), as follows:

$$PAR = 0.02092 * Rs * [1 - \exp(-0.65 * LAI)] \tag{2}$$

LAI is calculated according to the growth stage indicated by $Ihun$, if $Ihun < \lfloor dpl1 \rfloor * 0.01$,

$$LAI = \frac{(dpl1 - \lfloor dpl1 \rfloor) * Ihun}{\lfloor dpl1 \rfloor * 0.01} * blai \tag{3}$$

if $\lfloor dpl1 \rfloor * 0.01 \leq Ihun < \lfloor dpl2 \rfloor * 0.01$,

$$LAI = \left\{ (dpl1 - \lfloor dpl1 \rfloor) + \frac{[(dpl2 - \lfloor dpl2 \rfloor) - (dpl1 - \lfloor dpl1 \rfloor)] * (Ihun - \lfloor dpl1 \rfloor * 0.01)}{\lfloor dpl2 \rfloor * 0.01 - \lfloor dpl1 \rfloor * 0.01} \right\} * blai \tag{4}$$

if $\lfloor dpl2 \rfloor * 0.01 \leq Ihun < dlai$,

$$LAI = \left\{ (dpl2 - \lfloor dpl2 \rfloor) + \frac{[1 - (dpl2 - \lfloor dpl2 \rfloor)] * (Ihun - \lfloor dpl2 \rfloor * 0.01)}{dlai - \lfloor dpl2 \rfloor * 0.01} \right\} * blai \tag{5}$$

if $dlai < Ihun$,

$$LAI = 16 * blai\,(1 - Ihun)^2 \tag{6}$$

where $dlai$ is the fraction of growing season when growth declines, $dpl1$ and $dpl2$ are shape parameters of the LAI growth curve (see the definition in Table 1 in Ai et al., 2020), and $blai$ is the maximum leaf area index.

$REGF$ is calculated as:

$$REGF = \min(Ts, Ws, Ns, Ps) \tag{7}$$

where $Ts, Ws, Ns,$ and $Ps$ are the stress factors for temperature, water, nitrogen, and phosphorous, respectively. The details of

water and temperature stress are provided in the work of Ai et al. (2020). Nitrogen and phosphorous stress were not considered because of the lack of available information regarding fertilizer application (Hanasaki et al., 2008a).

The aboveground biomass ($Bag$) (kg ha$^{-1}$) is estimated with the accumulated biomass ($\sum \Delta B$) as:

$$Bag = [1 - (0.4 - 0.2 * Ihun)] \sum \Delta B \tag{8}$$

where $Ihun$ is the heat unit index, which is calculated as the ratio of accumulated daily heat units $\sum Huna(t)$ and the potential

heat unit ($Hun$):

$$Ihun = \frac{\sum Huna(t)}{Hun} \tag{9}$$



The daily heat units $Huna(t)$ is expressed as the difference between the daily mean air temperature ($T_a$) and the the crop's specific base temperature ($Tb$; provided as a crop-specific parameter):

$$Huna(t) = T_a - Tb \tag{10}$$

The crop yield ($Yld$) (kg ha$^{-1}$) is finally estimated from the aboveground biomass ($Bag$) using the crop-specific harvest index ($Harvest$) on the date of the harvest as:

$$Yld = Harvest * \frac{WSF}{WSF + \exp(6.117 - 0.086 * WSF)} * Bag \tag{11}$$

where $WSF$ is the ratio of $SWU$ (accumulated actual plant evapotranspiration in the second half of the growing season) to $SWP$ (accumulated potential evapotranspiration in the second half of the growing season):

$$WSF = \frac{SWU}{SWP} * 100 \tag{12}$$

Differences in crop type are expressed by the differences in crop parameters (e.g., $be$, $blai$, and $Tb$). Currently, the crop sub-model can simulate the yield for 18 food crops. The globally uniform default parameters for the food crops were collected from the default parameters of the SWIM model (Krysanova et al., 2000).

### 2.3 Algorithm improvement

Here, the crop sub-model was improved as follows. First, the effects of $CO_2$ fertilization and vapor pressure deficit change on radiation use efficiency were added to the H08 crop sub-model, using the equations and parameters adopted in SWAT (Neitsch et al., 2011; Arnold et al., 2013). In general, elevated $CO_2$ has a positive impact on crop yield, whereas increased vapor pressure deficit has a negative impact. The $CO_2$ fertilization effect is larger for $C_3$ crops (e.g., wheat, rice, and soybean) than for $C_4$ crops (e.g., maize). Specifically, the radiation use efficiency (be) is adjusted according to the concentration of $CO_2$ as:

$$be = \frac{100 * CO_2}{CO_2 + \exp(r_1 - r_2 * CO_2)} \tag{13}$$

where $be$ is the radiation use efficiency, $CO_2$ is the $CO_2$ concentration in the atmosphere (ppmv), $r_1$ and $r_2$ are shape coefficients.

$$r_1 = ln\left|\frac{CO_{2amb}}{0.01 * be_{amb}} - CO_{2amb}\right| + r_2 * CO_{2amb} \tag{14}$$

$$r_2 = \frac{ln\left|\frac{CO_{2amb}}{0.01 * be_{amb}} - CO_{2amb}\right| - ln\left|\frac{CO_{2hi}}{0.01 * be_{hi}} - CO_{2hi}\right|}{CO_{2hi} - CO_{2amb}} \tag{15}$$

where $CO_{2amb}$ is the ambient atmospheric $CO_2$ concentration (ppmv), $CO_{2hi}$ is an elevated atmospheric $CO_2$ concentration (ppmv), $be_{amb}$ is the $be$ of the crop at $CO_{2amb}$, and $be_{hi}$ is the $be$ of the crop at $CO_{2hi}$.

Additionally, the $be$ is adjusted with the vapor pressure deficit ($vpd$) (kPa) as:

$$be = be_{vpd=1} - \Delta be_{dcl} * (vpd - vpd_{thr}) \qquad \text{if } vpd > vpd_{thr} \tag{16}$$

$$be = be_{vpd=1} \qquad \text{if } vpd \le vpd_{thr} \tag{17}$$

where $be_{vpd=1}$ is the $be$ for the plant at a $vpd$ of 1 Kpa, $\Delta be_{dcl}$ is the rate of $be$ decline per unit increase in $vpd$, $vpd_{thr}$ is the threshold $vpd$ above which a plant will exhibit reduced radiation use efficiency. $vpd_{thr}$ is assumed to be 1 Kpa.

### 2.4 Parameter calibration

Next, we calibrated the key parameter of maximum leaf area index ($blai$) and adjusted harvest index ($Harvest$) accordingly by adopting the concept of management intensity in accordance with the method of Fader et al. (2010). Note that, for many



countries in the world, the historical annual crop yield from FAO shows an apparent increasing trend. Hence, the usual way of splitting data into two periods (i.e., former for calibration, the latter for validation) didn't work. Therefore, we used the mean of even years for calibration and the mean of odd years for confirmation. Specifically, we calibrated the maximum leaf area index by iterating the values from 0.5 to 7.1, with an interval of 0.3, under both rainfed and irrigation conditions in the even years from 1986 to 2015. The crop-specific best maximum leaf area index in each country was then determined as the value that can minimize the bias between the mean simulated yield and mean FAO statistical yield. When FAO statistical yield or simulated yield data are missing for a country, we took the original crop-specific default values. The calibration and confirmation results showed good agreement with the FAO statistics (Fig. S1).

### 2.5 Meteorological data

The ISIMIP3a GSWP3-W5E5 global meteorological data (available at https://data.isimip.org/search/tree/ISIMIP3a/InputData/climate/atmosphere/gswp3-w5e5/) from 1980 to 2015 were used in all simulations in this study. The spatial resolution of GSWP3-W5E5 data is 0.5°. Eight daily meteorological variables (downward shortwave radiation, downward longwave radiation, specific humidity, rainfall, snowfall, air pressure, wind speed, and air temperature) were used to run H08.

### 2.6 Reference yield data

To calibrate and validate the simulated crop yield, several yield data sets with different spatial resolutions were collected. The country-level yield data from FAO (available at https://www.fao.org/faostat/en/#data, final accessing date is May 9, 2022) and grid-level (0.5°) yield data from the Global Dataset of Historical Yield (GDHYv1.2+v1.3) (Iizumi et al., 2020) (available at https://doi.pangaea.de/10.1594/PANGAEA.909132) for the period of 1986 to 2015 were used to evaluate model performance. FAO statistical yield was reported as fresh matter, whereas the model simulated yield denotes the dry matter. For consistency in the comparisons, as reported by Farder et al. (2010) and Müller et al. (2017), the FAO statistical yield was converted to dry mater with a crop-specific factor (e.g., 0.88, 0.88, 0.87, and 0.91 for maize, wheat, rice, and soybean) in accordance with Wirsenius (2000). The global data set of historical yield for major crops (GDHY) yield data is a spatially explicit data set that converts the FAO annual national statistical yield to grid-level yield based on gridded net primary production estimated from several satellite products (Iizumi et al., 2020). The FAO statistical yield and GDHY yield provide valuable information for evaluation of crop model performances at country and grid levels, respectively (Müller et al., 2017; Iizumi et al., 2020).

### 2.7 Simulation setting and yield processing

After algorithm improvement and parameter optimization, two different simulations for maize, wheat, rice, and soybean were run under both rainfed and irrigation conditions from 1986 to 2015 on a daily scale. The simulation was performed with the default model and the improved model under the assumption that the four crops were planted and harvested in a hypothetical cropland of each grid cell. Under rainfed condition, the crop growth was subject to water stress; under irrigation condition, there was no effect of water stress on crop growth. The yield processing is as follows:

First, the gridded yield (*Yld*) was aggregated from simulated yield as follows:

$$Yld = \frac{Yld_{rain} \times Area_{rain} + Yld_{irri} \times Area_{irri}}{Area_{rain} + Area_{irri}}$$

where $Yld_{rain}$ and $Yld_{irri}$ are the simulated yield under rainfed and irrigation conditions, respectively. $Area_{rain}$ and $Area_{irri}$ are the rainfed and irrigated harvest area per crop in a grid cell, respectively. The rainfed and irrigated harvest areas per crop were obtained from MIRCA2000 data set (Portmann et al., 2010) (available at https://www.uni-frankfurt.de/45218031/Data_download_center_for_MIRCA2000).

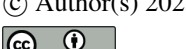



Then, the national yield was aggregated from the gridded yield and weighted according to the crop-specific total harvest area. Because reference yield data have limited quality for marginal and small areas (Müller et al., 2017), we considered grid cells with harvest area > 10 ha (Jägermeyr et al., 2021).

Finally, to ensure that the simulated data and reference data received similar treatment, we used the detrended yield when

comparing time series variations of simulated yield and reference yields (Müller et al., 2017). In accordance with the methods of previous studies (Müller et al., 2017; Iizumi et al., 2013; 2014a), the moving average method was used to remove the trends. Specifically, same as Müller et al. (2017), the anomaly yield was calculated by subtracting the moving average of a 5-year window.

**3 Results and discussion**

**3.1 Comparison with FAO statistical national yield**

Compared with the yield simulated by the default model, as shown in Fig. 1, the improved model showed better agreement with the FAO statistics of the mean national yield for the top 20 largest producer countries per crop (explaining approximately 88%, 86%, 93%, and 99% of global maize, wheat, rice, and soybean production, respectively).. First, the mean bias (difference

between mean national yield of simulation and mean national yield of FAO) of the 20 largest producer countries was considerably reduced to −4%, 3%, −1%, and 1% for maize, wheat, rice, and soybean, respectively. Second, the corresponding coefficients of determination ($R^2$) values of the mean national yield of simulation and the mean national yield of FAO increased from 0.01 to 0.98, 0.21 to 0.99, 0.06 to 0.99, and 0.14 to 0.97 for maize, wheat, rice, and soybean, respectively. Third, the corresponding root mean square error (RMSE) decreased from 7.1 to 1.1, 2.2 to 0.6, 2.7 to 0.5, 2.3 to 0.3 t/ha for maize, wheat,

rice, and soybean, respectively. These results suggested that the improved simulation could reliably reproduce the long-term averaged historical yield for the four major crops at the national level.

To investigate the capacity to reproduce the temporal variability of crop yield, a time series of detrended yield anomalies in simulation data and FAO data for the top 20 largest producer countries per crop are presented in Fig. 2 for maize and Figs.

S2–S4 for wheat, rice, and soybean, respectively. With regard to the ability to capture interannual variation in FAO yield, the model showed better performances for maize, wheat, and soybean than for rice. For example, positive correlations were found in 18, 16, 11, and 16 of the top 20 largest producer countries with the mean correlation coefficient (R) values of 0.48, 0.51, 0.31, and 0.36 for maize, wheat, rice, and soybean, respectively. The improved model showed better performance (increased R and decreased RMSE) than the default model particularly for maize and wheat.


The R and RMSE of time series detrended yield anomalies between simulated yield and FAO yield for the top five largest producer countries per crop are summarized in Fig. 3. These countries were selected to make the data comparable with the latest global crop model intercomparison study by Jägermeyr et al. (2021). Overall, the R and RMSE values of our simulations were within the range of current mainstream crop models reported by Jägermeyr et al. (2021). For maize, wheat, and soybean,

the R and RMSE values of our simulation were comparable with the ensemble means of different crop models reported by Jägermeyr et al. (2021); for rice, our simulation showed higher R values (except in Bangladesh and China) and lower RMSE values. However, the metric scores of our improved model and the other crop models in the work of Jägermeyr et al. (2021) remained low (e.g., few countries had R values > 0.5). This finding suggested that current crop models continue to experience difficulty in fully capturing the interannual variation of the historical yield because crop models only reflect the interannual

climate signals in the simulated yields (Jägermeyr et al., 2021). This also indirectly implied that the climate variation might not be the main driver of the interannual yield variation for the major producer countries.



To further validate the above conjecture, we investigated the impacts of climate variables (i.e., precipitation and air temperature) on interannual yield variation by analyzing the correlations of total precipitation/mean air temperature in the growing season with the annual yield per crop. Using maize as an example (Fig. 4), there were no statistically significant relationships ($p > 0.05$) between precipitation and FAO statistical yield for most of the top 20 largest producer countries (17/20). Significant positive correlations between precipitation and the FAO statistical yield ($p < 0.05$) were found in only three countries: Romania, Hungary, and Serbia. The crop yield estimation relies on water availability; therefore, the variation in yield simulation largely reflects variation in precipitation. Accordingly, we observed good simulation performance in those three countries (Fig. 2) with a clear correlation between FAO yield and precipitation (Fig. 4). Also, there were no statistically significant relationships between air temperature and FAO statistical yield for most of the top 20 largest producer countries (12/20) (Fig. 5). Similarly, there were no statistically significant correlations between precipitation/air temperature and FAO statistical yield in most countries for wheat, rice, and soybean (see Supplementary Figs. S5−10).

## 3.2 Comparison with GDHY gridded yield

Spatially explicit yield data enabled us to more fully evaluate the spatial distribution of model simulations. We compared the spatial distribution between simulated crop yield (before and after improvement) and the GDHY yield data set. Using maize as an example, apparent overestimation was detected in many parts of the world (e.g., China, Argentina, Brazil, India, Indonesia, Thailand, Mexico, and most countries in Africa) in the default simulation (Fig. 6a). In contrast, the improved simulation (Fig. 6b) showed a spatial pattern similar to the GDHY yield data (Fig. 6c). For the yields of wheat, rice, and soybean, the spatial distribution after improvement also showed a pattern similar to the GDHY yield data (Supplementary Figs. S11−13).

In accordance with the method of Müller et al. (2017), we conducted grid-level time series analysis of the correlations of the detrended yield between simulated and GDHY data (Fig. 7) to further identify the differences in the two yield data sets. Using maize as an example (Fig. 7a), statistically significant correlations ($p < 0.1$) were observed in a wide of range of regions (e.g., northeast USA, southern Europe, northeastern China, southern Brazil, eastern Argentina, southern Africa, and eastern Australia) (Fig. 7a). Notably, there were also substantial differences in a considerable number of locations without statistically significant correlations ($p > 0.1$) (e.g., southeastern USA, western and central Asia, Brazil, and central Africa) (Fig. 7a). Similar characteristics were found for wheat, rice, and soybean (Fig. 7b–d).

Such similarities or discrepancies between two yield data sets have been observed previously (see Fig. 9 in Müller et al., 2017). For example, there were statistically significant correlations ($p < 0.1$) and no statistically significant correlations ($p > 0.1$) between two data sets developed by Iizumi et al. (2014b; an earlier version of GDHY used in this study) and Ray et al. (2012) in a wide of regions. Such comparisons can help to identify considerable disagreements in global estimates of the spatial distribution of crop yield (Kim et al., 2021). Because it is difficult to determine whether one of these estimates is better than the others, the disagreement between our simulation and the GDHY data does not necessarily indicate that our simulation quality is low.

## 3.3 Limitations

Although crop yield simulations were improved, there were several limitations because of the assumptions, methods, and data sets used in this study. First, in accordance with the methods of previous studies (Müller et al., 2017; Jägermeyr et al., 2021), yield calculation and aggregation were conducted with the assumption that the irrigated harvest area and total harvest area per crop did not change throughout the study period; this assumption was based on data availability. However, these aspects do





change over time. To overcome the problems associated with such an assumption, dynamic harvest area data at annul intervals should be developed in future studies. Second, our calibration was conducted at the national scale in accordance with the method of Fader et al. (2010), rather than using finer spatial scale (e.g., subnational or gird-level), which increased the uncertainty of the yield simulations within each country. As shown in Figure 6, the yield distribution is highly variable within a specific country. To incorporate the spatial heterogeneity in crop yield, ideally, parameter calibration should be conducted at grid-cell level (e.g., Iizumi et al., 2009; Sakurai et al., 2014). Although this approach has long-term promise, it is technically challenging because of uncertainty in the global gridded yield products and the potential for inflation in the parameter optimization calculation. In addition, the calibrated parameter reflected the mean average sate, therefore might ignore the year-by-year variation. Third, the reference data set from GDHY does not represent purely observation-based yield, therefore, it is subject to errors or uncertainty resulting from its own methodology (e.g., errors in gross primary production and crop stress response) (Müller et al., 2017). Nonetheless, at current stage, both FAO and GDHY data sets remain good references for evaluating the performances of crop models, as suggested or widely used in previous studies (Müller et al., 2017; Iizumi et al., 2020; Jägermeyr et al., 2021). Finally, our crop model is a simple model that does not fully represent the factors influencing crop growth. For example, we did not explicitly simulate N and P processes, although these effects are now reflected in the calibrated parameters (Fader et al., 2010). Additionally, the waterlogging effect is underrepresented in most crop models, including our model (Jägermeyr et al., 2021). Such physical mechanisms should be addressed in the development of future models.

## 4 Case study to estimate the contribution of irrigation to global food production

Finally, to demonstrate the improved model can be applied for various food-water nexus study, a well-recognized study by Döll and Sibert (2010) which estimated the contribution of irrigation on global food production is revisited and traced. To trace their work, a global crop yield model is needed which is capable to estimate crop yield reasonably well and deal with the effect of irrigation explicitly.

Irrigation plays a critical role in global food production. The literature usually indicates that approximately 40% of global total food production is from irrigated land (Postel et al., 2001; Siebert et al., 2005; Abdullah et al., 2006; Khan et al., 2006; Wada et al., 2013; Perrone et al., 2020; Ringler et al., 2020; Borsato et al., 2020), but the rationale and country-specific variation have not been fully explained. To our knowledge, Postel (1992) reported one of the first estimates, whereby approximately 36% of the global food production was from irrigated land based on statistical data. Then, Siebert and Doll (2010) reported that irrigation contributed to approximately 33% of the global total production. Here, we revisited the irrigation contributions for global production of maize, wheat, rice, and soybean using our improved model. Irrigation contribution in percentage ($I$) in a country ($c$) is defined as: $I,c = \frac{Yld_{irri,c}*Area_{irri,c}}{Yld_{irri,c}*Area_{irri,c}+Yld_{rain,c}*Area_{rain,c}}*100\%$, where $Yld_{irri,c}$ and $Yld_{rain},c$ are the irrigated and rainfed yields for a country, respectively; $Area_{irri,c}$ and $Area_{rain,c}$ are the total irrigated and rainfed harvest areas for a country, respectively.

Our results showed that the global average production levels from irrigated cropland were approximately 27%, 30%, 61%, and 16% for maize, wheat, rice, and soybean, respectively (Fig. 8). These estimates were close to the estimates of Siebert and Doll (2010): 26%, 37%, 77%, and 8%, respectively. The similarities between these two studies mainly arose because both studies used data from Portmann et al. (2010) for crop-specific harvested area, and both models were calibrated with FAO data.

## 5 Conclusions



In this study, we improved the capacity of H08 to simulate the yields of four major staple crops: maize, wheat, rice, and soybean. The improved national yield estimates generally showed good consistency with FAO statistical national yields. The improved grid-level yield estimates showed similarities in terms of spatial patterns and the reproduction of interannual variation, compared with GDHY yield over a wide area, although there were substantial differences in other places. As reported

in previous studies, the full reproduction of historical interannual yield variation remains challenging for global gridded crop modelling. Finally, we quantified the contributions of irrigation to the global production of maize, wheat, rice, and soybean; we explored the variations in irrigation contributions among countries. Together with the ability to simulate bioenergy crop yield (Ai et al., 2020; 2021), to our knowledge, our improvements provide a good tool that can simultaneously simulate bioenergy potential and crop production while specifying irrigation water withdrawal into the most detailed sources within a

single framework, which will be beneficial for advancing global food–water–energy–land nexus studies in the future (e.g., planetary boundary, virtual water trade, and sustainable development goals).

*Code and data availability.* The mode code used here is archived on Zenodo (https://zenodo.org/record/7344809#.Y3xnU7JBzjA). Technical information regarding H08 model is available from:

https://h08.nies.go.jp/h08/. The links to the data sets used in this study are provided in the main text.

*Competing interests.* The authors declare that they have no conflict of interest.

*Author contribution.* Zhipin Ai and Naota Hanasaki designed this study. Zhipin Ai collected the data, developed the model

code, and performed the simulations. Zhipin Ai and Naota Hanasaki wrote the manuscript.

*Acknowledgments.* This study was supported by the Environment Research and Technology Development Fund (JPMEERF20202005) of the Environmental Restoration and Conservation Agency, Japan. We'd like to acknowledge Dr. Kazuya Nishina and Dr. Masahi Okada for their great suggestions.



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



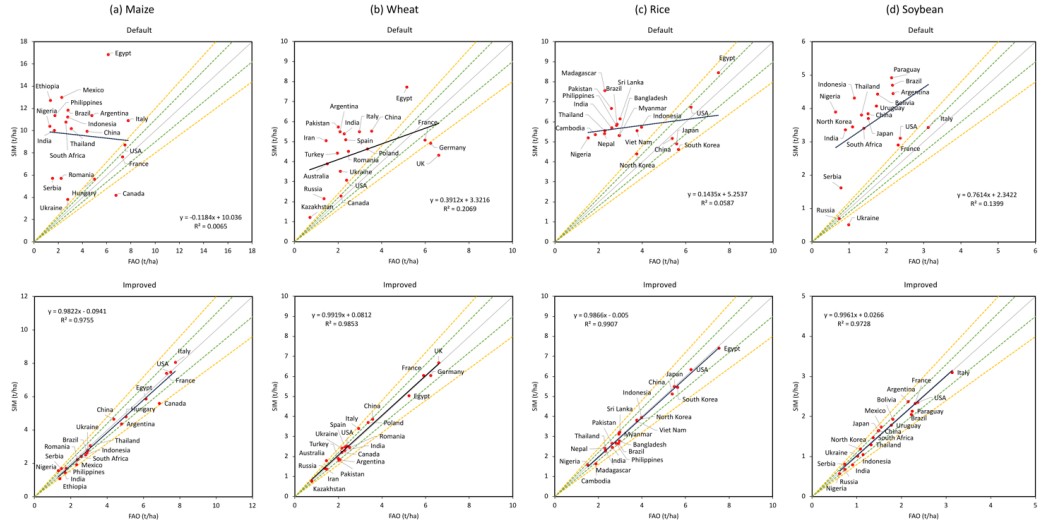


**Fig. 1 Comparison of mean simulated yield and mean FAO yield for the top 20 largest producer countries from 1986 to 2015. Default and improved indicates simulation using the default and improved model, respectively. Dashed green and yellow lines indicate ±10% and ±20% differences, respectively. SIM denotes simulated yield and FAO denotes reported yield from FAO. Panel (a) for maize, (b) for wheat, (c) for rice, and (d) for soybean, respectively.**




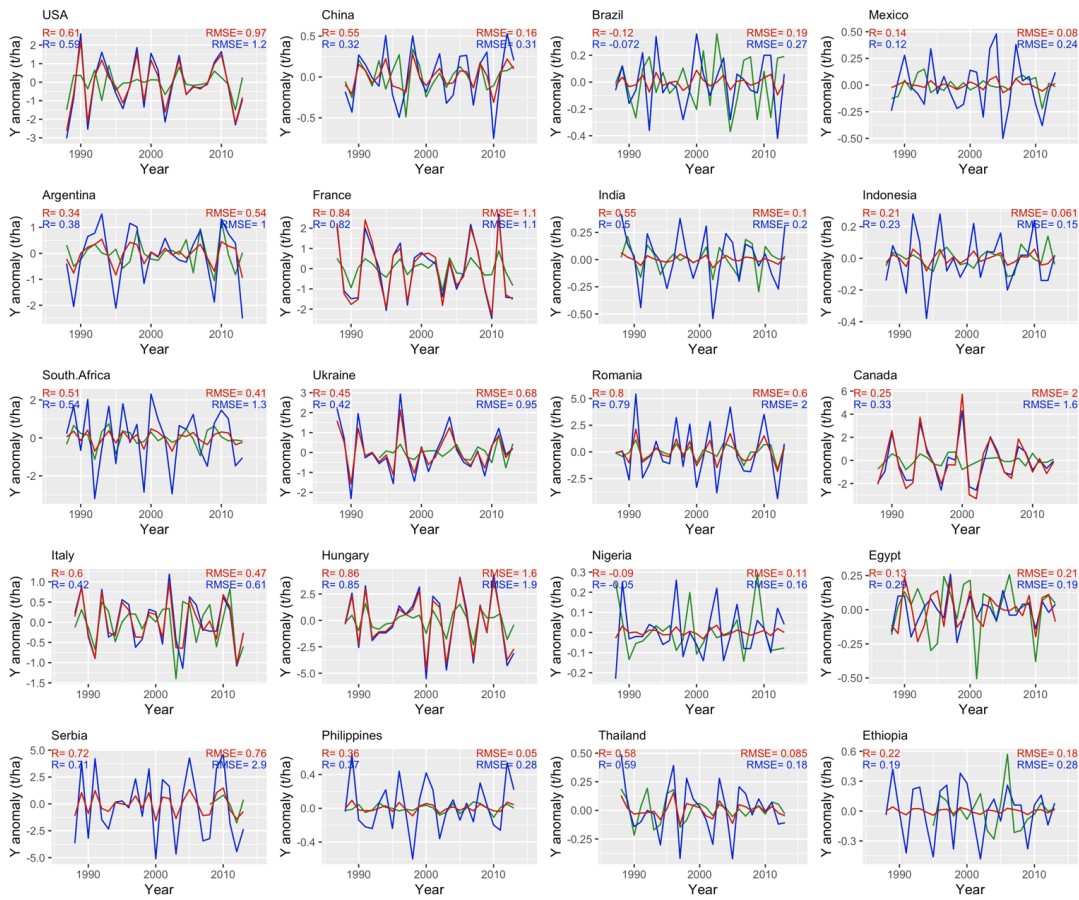

**Fig. 2 Time series detrended maize yield anomalies from improved simulation (red), default simulation (blue), and FAO (green) for the top 20 largest producer countries. Y, yield; R, correlation coefficient; RMSE, root mean square error.**





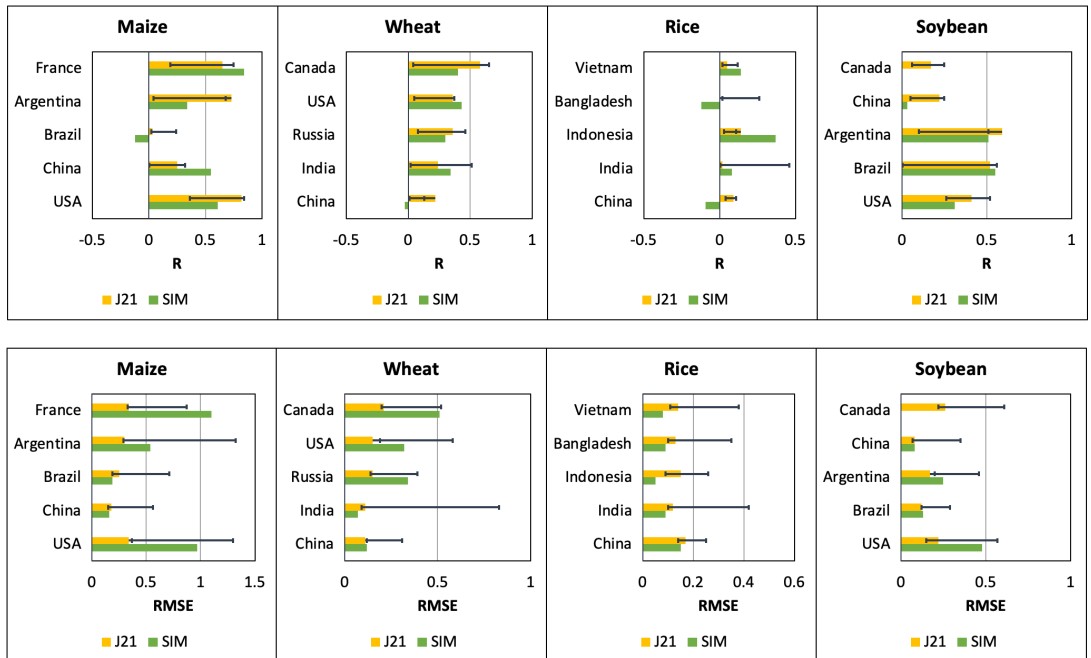

**Fig. 3 Comparison of R and RMSE values of time series detrended yield anomalies between this study (SIM) and Jägermeyr et al. (2021) (J21). Yellow bar denotes ensemble mean of different crop models used in the work of Jägermeyr et al. (2021). Error bars indicate maximum and minimum values among different crop models.**



**Fig. 4 Relationship between maize yield (blue: simulated; red: FAO) and total precipitation in the growing season from 1986 to 2015 for the top 20 largest producer countries. Y, yield; P, precipitation; R, correlation coefficient.**





**Fig. 5 Relationship between maize yield (blue: simulated; red: FAO) and mean air temperature in the growing season from 1986 to 2015 for the top 20 largest producer countries. Y, yield, T, air temperature; R, correlation coefficient.**



**(a)**

**(b)**

**(c)**

Fig. 6 Spatial distribution of the mean (1986-2015) simulated yield (a, default; b, improved) and GDHY yield (c) of
maize. Units in the legend are t/ha.



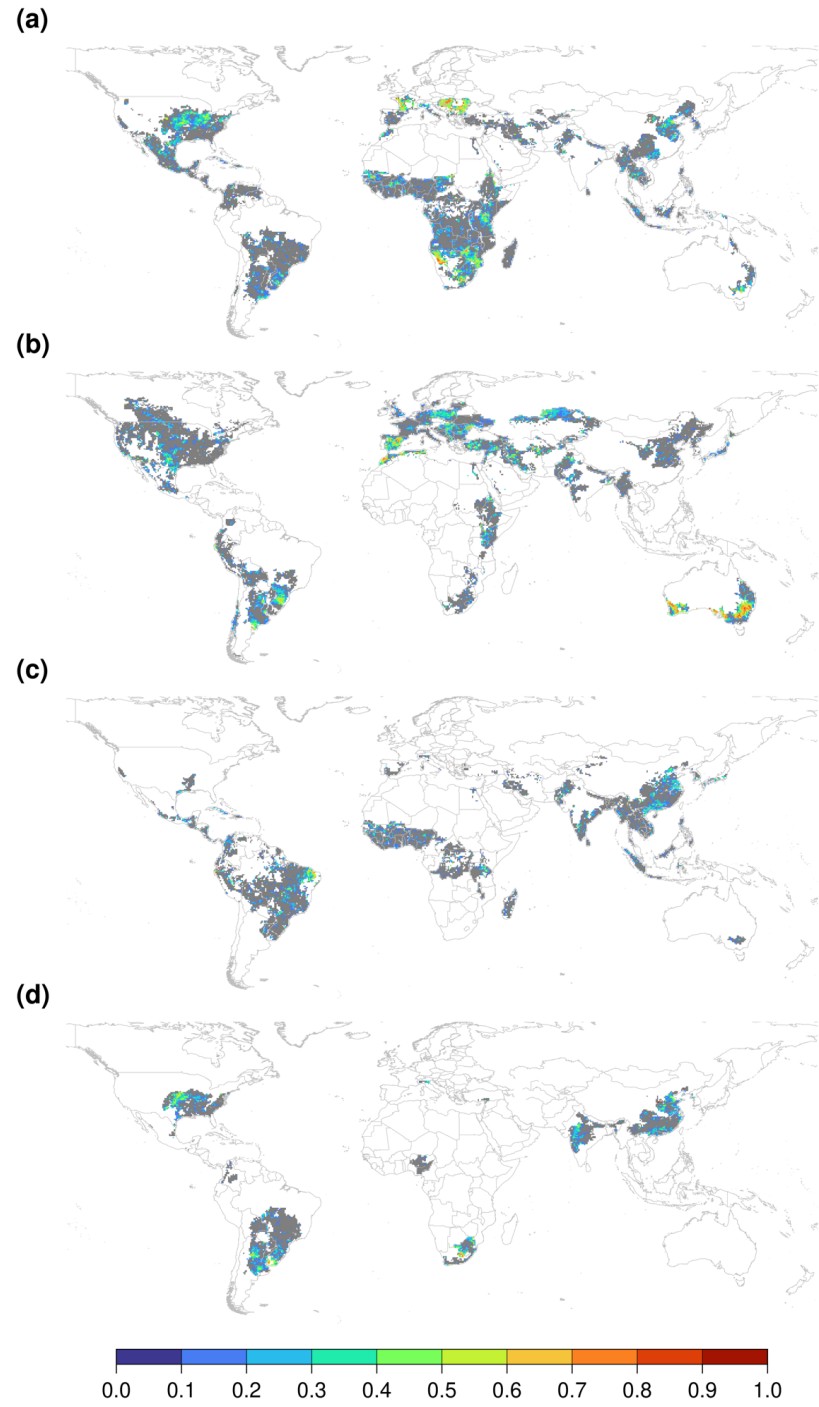

**Fig. 7 Time series correlation between simulated yield and GDHY yield after trend removal using a 5-year moving average. Gray areas indicate no statistically significant correlation between the two data sets (p > 0.1), and white areas indicate no yield data for that crop in at least one of the two data sets. Panel (a) shows determination coefficient for maize, (b) for wheat, (c) for rice, (d) for soybean, respectively.**




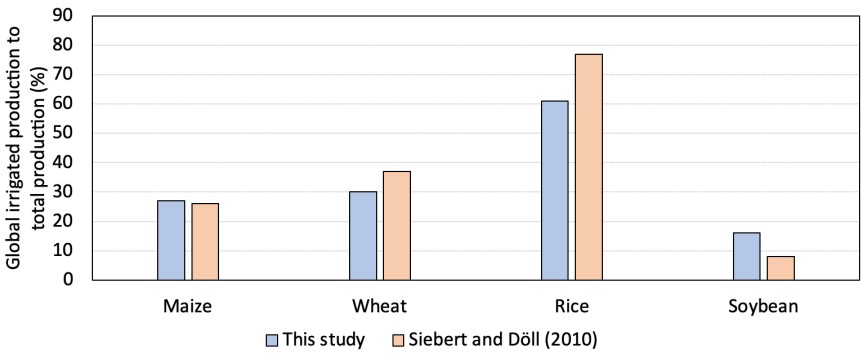

**Fig. 8 Percentages of irrigation contribution to the global production of maize, wheat, rice, and soybean, respectively.**