# Peer review of "Simulation of crop yield using the global hydrological model H08 (crp.v1)"

_Geoscientific Model Development, 2022_

## Author Comment (AC1)

**Response to Reviewer 1**

**R1C1:** In their submitted paper 'Simulation of crop yield using the global hydrological model H08 (crp.v1)', the authors enhance the H08 crop sub-model with parameter calibration and algorithm improvement. Thereby, the $CO_2$ fertilization effect and the effect of vapor pressure deficit change has been included to the model. Additionally, a model calibration has been applied. In order to evaluate the model results, simulated yields are compared with statistical yields and other global crop models for the major 4 crops (maize, wheat, rice, soybean) at country and grid-level. The paper is well written and understandable. Nevertheless, the paper has a main weakness: If you calibrate your model towards yields that you also use to validate/evaluate your model, it's not a surprise that $R^2$ is >0.99. But does that mean that your model improved? I wouldn't say so. It just says that the calibration was successful.

**Response:** We thank the reviewer for the constructive comments. We thoroughly revised unclear and incomplete points and incorporated all remarks.

Specifically, we conducted new simulations (see Table 1 and Fig. 1 for details) and added the results to explain the effects of $CO_2$ fertilization, vapor pressure deficit, and the combined effects of $CO_2$ fertilization and vapor pressure deficit on yield and crop water productivity before model calibration. Details are given in section 3.1 (see response R1C3).

Table 1. Simulations setting

| Simulation ID | CO2 effect | VPD effect | Calibration |
|---|---|---|---|
| D | No | No | No |
| C | Yes | No | No |
| V | No | Yes | No |
| CV | Yes | Yes | No |
| CVC | Yes | Yes | Yes |

We have also modified relevant text by replacing the "improved simulations" with "calibrated simulations" as the comment said in R1C2.

[Figure]

*Fig. 1 Comparison of the mean yield from 1986 to 2015 of different simulations and FAO statistics. (a) maize, (b) wheat, (c) rice, and (d) soybean. Further details on five utilized simulations (D, C, V, CV, and CVC) are listed in Table 1.*

**R1C2:** First, I'd suggest to say 'calibrated simulations' and 'default simulations' instead of 'improved' and 'default' simulations throughout the manuscript.

**Response:** We replaced the "improved simulations" with "calibrated simulations"

**R1C3:** Second, given the fact that you added the effects of $CO_2$ and vapor pressure deficit to the H08 model, it would be interesting in this study to quantify the difference between considering these effects and not. As in Deryng et al. (2016) I would encourage you to quantify the difference of $CO_2$ effect on crop water productivity for C3 and C4 crops.

**Response:** We added the requested results to explain the effects of $CO_2$ fertilization, vapor pressure deficit, and the combined effects of $CO_2$ fertilization and vapor pressure deficit on the yield (Fig. 1) and crop water productivity (Fig. S1):

*"3.1 Effects of CO2 fertilization and vapor pressure deficit*

*When only considering the $CO_2$ fertilization effect (simulation C), there was a positive impact on crop yield, as compared to default simulations (simulation D) (Fig. 1). In addition, similar to previous studies (e.g., Deryng et a., 2016), the $CO_2$ fertilization effect is larger for C3 crops (wheat, rice, and soybean) than for C4 crops (maize). In contrast, when only considering the vapor pressure deficit effect (simulation V), there was a negative impact on crop yield in comparison with default simulation. When considering the effects of both CO2 fertilization and vapor pressure deficit, there was a positive impact on crop yield for the majority of the top 20 largest producer countries, while a negative impact was found for some countries (e.g., India and Egypt for maize). These impacts were also reflected in crop water productivity (CWP, defined as the ratio of crop yield to evapotranspiration). The averaged change of CWP in the top 20 largest producer countries was 4.8%, −2.3%, and 2.5% for maize under simulations C, V, and CV, compared to simulation D (Fig. S2). The corresponding values were (6.4%, −1.1%, 5.3%), (5.8%, −3.4%, 2.3%), and (7.1%, −3.6%, 3.4%) for wheat, rice, and soybean, respectively."*

**R1C4:** A calibration can be done in a next step but after the validation. The quantification of water flows would require to validate crop evapotranspiration, which is not done in this study. Given the contextual and structural deficits in this study, I'd suggest major revisions.

**Response:** We revised the manuscript and show the effects of $CO_2$ fertilization and vapor pressure deficit on crop yield in section 3.1 and calibration results in section 3.2. The main objective of this study was to calibrate the model for crop yield and to add two new functions to determine the impact of $CO_2$ fertilization and vapor pressure deficit change on crop yield. Regarding crop evapotranspiration, a comprehensive analysis of blue water, green water, and crop evapotranspiration has been performed in an earlier study by Hanasaki et al., (2010). Note that the validity of overall hydrology is thoroughly discussed in Hanasaki et al. (2018).

Specific Comments:

**R1C5:** Abstract ln 1: 'Food and water are essential for life'. To me, this is trivial and a bit pathethic.

**Response:** We removed this sentence.

**R1C6:** Abstract ln 19: What means 'reasonably'? Can you quantify that with a statistical value?

**Response:** Detailed scientific discussions of case study results are not permitted in this journal, as noted by the editor in the initial editor review, and we have therefore revised this sentence. It now reads as follows (Lines 16–18):

*"Using the enhanced model, we quantified the contributions of irrigation to global food production and compared our results to an earlier study."*

**R1C7:** Line 35: You could add the PROMET model to this list of models, because it is also a hydrological model with an enhanced crop growth module included that has been applied at global (Zabel et al. 2019) and regional scale (Degife et al. 2021).

**Response:** Thank you. We added the PROMET model.

**R1C8:** Line 76: Is it still a 'process-based model', after calibrating parameters to match statistical yields that are subsequently used for model evaluation?

**Response:** We were referring to the original method of Fader et al. (2010), which was also implemented within the global, process-based model LPJmL. As the reviewer clarified, our model is not a rigorous "process-based model" after calibration. Therefore, we removed "process-based model", and the sentence now reads as follows (Lines 75–77):

*"Then, we adopted the method of Fader et al. (2010) for parameter calibration because of its robust performance, minimal computation costs, simplicity of implementation, and because the method requires only national yield data which are easily accessible and generally reliable."*

**R1C9:** Line 125: Nitrogen and phosphorous stress are implicitly considered in your calibration procedure! It possibly one of the main factors that influences your calibration.

**Response:** We agree and revised the sentence (Lines 138–140):

*"Nitrogen and phosphorous stress were not considered in the original model (Hanasaki et al., 2008a) and were indirectly represented in the calibration simulation in the present study."*

**R1C10:** Line 224: Two full stops.

**Response:** Thank you. We deleted one full stop.

**R1C11:** Line 222-231: See above comment on calibration and validation.

**Response:** We replaced "improved simulations" with "calibrated simulations".

**R1C12:** Line 233-239: Interannual yield variabilities are much higher in both, calibrated and default simulations than in observations. Can you explain why?

**Response:** Using maize as an example (Fig. 3), the yield anomaly magnitude becomes closer to FAO data for the majority of the top 20 countries after calibration. We noted that, for some countries, including USA, France, Ukraine, and Canada, the yield anomalies were still higher compared to FAO data, which is likely to be because the default simulations were already comparable to the FAO data and the calibration resulted in limited improvement (Fig. 1a and Fig. 2a). Therefore, the anomaly magnitudes in these countries did not improve much after calibration. We added these results in Lines 258–261:

*"Note that the calibrated model showed a similar performance to that of the default model in some countries (e.g., in USA, France, Ukraine, and Canada for maize) because the default simulations were already comparable to yield reported by the FAO, meaning that the calibration resulted in limited improvement (see Figs. 1a and 2a)."*

**R1C13:** Line 243: According to the GGCMI phase 3 protocol, none of the models in Jägermeyr et al. (2021) are calibrated to yields.

**Response:** We agree that the GGCMI participating models were not specifically calibrated for the protocol of GGCMI 3. However, at least for some models, the model description papers indicate that their parameters were calibrated during development. For example, for the model CROVER used in Jägermeyr et al. (2021), the crop parameters were calibrated using the GDHY yield dataset at the grid-cell level (Okada et al., 2018; Okada et al., 2015). In addition, for LPJmL model, the parameters were calibrated to FAO data (Bondeau et al., 2007; Fader et al., 2010).

**R1C14:** Line 295: Same problem than with constant irrigation occurs for the crop calendar, I guess?

**Response:** The crop calendar used in our simulation varies yearly based on differences in meteorological inputs. The crop calendar for H08 was determined as follows: first, the crop sub-model calculated harvesting date and yield for a crop planted on every day from January 1 to December 31. If the air temperature during the period dropped below the temperature threshold for cold death, crops would have perished and yield would have been zero. The planting date that resulted in the greatest yield over the year was assumed to be the planting date for the tested crop and location, and the yield for that date is represented the potential yield.

**R1C15:** Line 305: What means 'remain good references'? Don't you contradict yourself with what is said before? Of course, we need better data and this is a strong limitation.

**Response:** Thank you. We removed the sentence.

**R1C16:** Line 307: 'factors' or better say 'processes' here.

**Response:** Thank you. We replaced "factors" with "processes".

**R1C17:** Line 317: Therefore, it would be required to validate simulated crop evapotranspiration. If you can reproduce yields, it does not mean that evapotranspiration is simulated correctly.

**Response:** We focused on crop yield calibration and did not consider hydrological modelling, including evapotranspiration. The model outputs on bule water, green water, and total evapotranspiration were analyzed in an earlier study (Hanasaki et al., 2010) during the development of H08. In addition, the validity of overall hydrology was thoroughly discussed in Hanasaki et al. (2018).

**R1C17:** Line 331: You just mentioned that irrigated areas are kept constant over time, which impacts the results. Another question is the amount of irrigation that is applied. Do you consider only full irrigation, or do you also consider deficit irrigation?

**Response:** Thank you. We only consider full irrigation.

**R1C18:** Line 336: As I understood, H08 had the capacity to simulate yields before. You added two processes and applied a calibration.

**Response:** Thank you. we revised the sentence (Lines 360–361):

*"In this study, we determined the effects of $CO_2$ fertilization and vapor pressure deficit on crop yield using the global hydrological model H08. Then, we calibrated the yields of four major staple crops: maize, wheat, rice, and soybean."*

**R1C19:** Line 343: Please avoid qualitative statements like 'a good tool'. Nobody can say what is a good tool.

**Response:** We removed "good".

**R1C20:** Figure 1: Font size too small. Country names, etc. are very hard to read, even after zooming in.

**Response:** Thank you. We revised the figure and increased the font size.

**R1C21:** Figure 3: Interesting to see R and RMSE values as a bar plot, which is not intuitive. I'd suggest to show a Taylor diagram or a scatterplot.

**Response:** Thank you. We understand that Taylor diagram is able to simultaneously present the R and RMSE in one figure. However, this is not the main purpose here, as our intention was to compare the calculated metrics (R and RMSE) from our study with those reported in Fig. S10 in Jägermeyr et al. (2021). In addition, to generate a Taylor diagram, we require the data of both observation and simulation, but the original data used in Fig. S10 in Jägermeyr et al. (2021) are not available. Therefore, we chose to present the metric score comparison in a bar plot. We hope you understand.

**References:**

Bondeau, A., Smith, P. C., Zaehle, S., Schaphoff, S., Lucht, W., Cramer, W., Gerten, D., Lotze-Campen, H., Müller, C., Reichstein, M., and Smith, B.: Modelling the role of agriculture for the 20th century global terrestrial carbon balance, Glob. Change Biol., 13, 679–706, https://doi:10.1111/j.1365-2486.2006.01305.x, 2007.

Fader, M., Rost, S., Müller, C., Bondeau, A., and Gerten, D.: Virtual water content of temperate cereals and maize: Present and potential future patterns. J. Hydrol., 384(3–4), 218–231, https://doi.org/10.1016/j.jhydrol.2009.12.011, 2010.

Hanasaki, N., Kanae, S., Oki, T., Masuda, K., Motoya, K., Shirakawa, N., Shen, Y., and Tanaka, K.: An integrated model for the assessment of global water resources – Part 1: Model description and input meteorological forcing, Hydrol. Earth Syst. Sci., 12, 1007–1025, https://doi:10.5194/hess-12-1007-2008, 2008a.

Hanasaki, N., Inuzuka, T., Kanae, S., and Oki, T.: An estimation of global virtual water flow and sources of water withdrawal for major crops and livestock products using a global hydrological model. Journal of Hydrology, 384(3-4), 232-244, https://doi:10.1016/j.jhydrol.2009.09.028, 2010.

Hanasaki, N., Yoshikawa, S., Pokhrel, Y., and Kanae, S.: A global hydrological simulation to specify the sources of water used by humans, Hydrol. Earth Syst. Sci., 22, 789–817, https://doi:10.5194/hess-22-789-2018, 2018.

Jägermeyr, J., Müller, C., Ruane, A. C., Elliott, J., Balkovic, J., Castillo, O., Faye, B., Foster, I., Folberth, C., Franke, J., Fuchs, K., Guarin, J., Heinke, J., Hoogenboom, G., Iizumi, T., Jain, A., Kelly, D., Khabarow, N., Lange, S., Lin, T., Liu, W., Mialyk, O., Minoli, S., Moyer, E., Okada, M., Phillips, M., Porter, C., Rabin, S., Scheer, C., Schneider, J., Schyns, J., Skalsky, R., Smerald, A., Stella, T., Stephens, H., Webber, H., Zabel, F., and Rosenzweig, C.: Climate impacts on global agriculture emerge earlier in new generation of climate and crop models. Nat. Food., 2(11), 873–885, https://doi.org/10.1038/s43016-021-00400-y, 2021.

Okada, M., Iizumi, T., Sakurai, G., Hanasaki, N., Sakai, T., Okamoto, K., and Yokozawa, M.: Modeling irrigation-based climate change adaptation in agriculture: Model development and evaluation in Northeast China. J. Adv. Model. Earth Sy., 7(3), 1409–1424, https://doi.org/10.1002/2014MS000402, 2015.

Okada, M., Iizumi, T., Sakamoto, T., Kotoku, M., Sakurai, G., Hijioka, Y., and Nishimori, M.: Varying benefits of irrigation expansion for crop production under a changing climate and competitive water use among crops. Earth's Future, 6(9), 1207–1220, https://doi.org/10.1029/2017EF000763, 2018.

---

## Author Comment (AC2)

**Response to Reviewer 2**

**R2C1:** This paper attempts to improve the capacity of a global hydrological model H08 to simulate the yields of four major crops. The authors substantially improve the crop growth algorithm and add additional calibration procedures to aim at a better fit with national crop yields reported by FAO. The latter is arguably the main point of concern here. Calibration and validation were performed using two very similar reference datasets, both based on the long-term mean of FAO's yields. Due to this similarity, the validation shows great agreement with reference data. However, once the calibrated model is compared to long-term (not mean) annual yields, the agreement substantially drops and hence the model provides a weak representation of the actual historical yields. Another issue lies in the insufficient description of Materials and Methods. As a reader, it is difficult to understand the details of crop growth processes, simulation setup, and calibration procedure. Therefore, this section would greatly benefit from adding figures (e.g. crop sub-model scheme) and tables (e.g. main input data). Lastly, it would be interesting to see if the improved model can simulate the water cycle more accurately since H08 is primarily a hydrological model. Given mentioned-above, I would suggest major revisions.

**Response:** We thank the reviewer for the constructive comments. We have thoroughly revised unclear and incomplete points and incorporated all remarks. We address your general comments in view of three aspects:

**1. Regarding the calibration**

We believe it is successful in reproducing the long-term average crop yield (Figs. 1 and 2) and capturing the interannual yield variation (Fig. 3). For the majority of the top 20 countries, the anomaly magnitude of the calibrated simulations became closer to anomalies based on FAO data. We added text in the abstract (Lines 13–15) and the results section (Lines 261–264) to acknowledge the weakness of our current model in fully capturing the interannual variation of the historical yield:

Lines 13–15:

"*The capacity of our model to capture the interannual yield variability observed in FAO yield was limited, although the performance of our model was comparable with that of other mainstream global crop models.*"

Lines 261–264:

"*Note that the calibrated model showed a similar performance to that of the default model in some countries (e.g., in USA, France, Ukraine, and Canada for maize) because the default simulations were already comparable to yield reported by the FAO, meaning that the calibration resulted in limited improvement (see Figs. 1a and 2a).*"

**2. Regarding the description of materials and methods**

We added tables to specify the simulation settings (Table 1), adjustment of the harvest index (Table S1), and the main input datasets (Table S2).

We also added a reference to the schematic figure showing the biophysical process of the crop sub-model (Lines 108–109):

*"A schematic figure that shows the basic biophysical processes of the crop sub-model is shown in Fig. 1b in Ai et al. (2020)."*

**3. Regarding the water cycle**

There will be no change for the water cycle since we did not modify the hydrological modelling. Instead, as we repotted here, our enhancement can help to simulate the crop yield and food production more accurately. Therefore, our work is important to realize hydrology and crop growth coupled model to advance so-called food-energy-water nexus studies. Our model would be used to quantify the tradeoff between irrigation and crop production (e.g., Ai et al., 2021; Heck et al., 2020) and advance the analysis of water embedded in agricultural products and their international trade (e.g., Hanasaki et al. 2010; Dalin et al. 2017).

**Specific comments**

**R2C2:** Title: What does (crp.v1) mean? do you really need to include it in a title?

**Response:** As required by the journal, we need to specify the model version. This abbreviation is used to distinguish the model version.

**R2C3:** Line 9-14: The authors put too much emphasis on the "good consistency" (how do you define good?) with the long-term mean of FAO's yields. However, they fail to acknowledge the poor correlation when yields were analysed not as a mean but as a time series over the same period. The latter, one could argue, is much more important in the context of solving water-food nexus problems.

**Response:** Thank you. We agree and revised the text by further clarifying "good consistency" in terms of reproducing long-term averaged yield while also acknowledging limitations in terms of capturing the interannual yield variation (Lines 11–15):

*"The simulated yield showed good consistency with FAO national yield. The mean biases of the major producer countries were considerably reduced to 2%, 2%, −2%, and −1% for maize, wheat, rice, and soybean, respectively. The capacity of our model to capture the interannual yield variability observed in FAO yield was limited, although the performance of our model was comparable with that of other mainstream global crop models."*

**R2C4:** Line 20: Why do you refer to "food-water-land-energy nexus"? so far you only talk about water and food. Similar thing for line 345.

**Response:** Thank you. We changed "food-water-land-energy nexus" to "food-water". In our previous work, we worked on estimating energy crop for a truly "food-water-land-energy" nexus study (Ai et al. 2021), but we agree with you that we should focus on "food-water" in this study.

**R2C5:** Line 35: If you are trying to list all global crop models, then add the ones you miss from the recent round of ISIMIP simulations, e.g. ACEA, PROMET, SIMPLACE etc. (Jägermeyr et al., 2021).

**Response:** Thank you. We added the three modes, it now reads as follows (Lines 34–38):

*"Many models have successfully incorporated the crop growth process and can simulate the global crop yield. These include LPJmL (Bondeau et al., 2007; Fader et al., 2010), GEPIC (Liu et al., 2007), PROMET (Mauser and Bach, 2009), PEGASUS (Deryng et al., 2011), CLM-Crop (Drewniak et al., 2013), PRYSBI2 (Sakurai et al., 2014), pAPSIM (Elliott et al., 2014), pDSSAT (Elliott et al., 2014), CROVER (Okada et al., 2015), ORCHIDEE-crop (Wu et al., 2016), PEPIC (Liu et al., 2016), MATCRO (Masutomi et al., 2016), SIMPLACE-LINTUL5 (Webber et al., 2016), and ACEA (Mialyk et al., 2022)."*

**R2C6:** Lines 108-162: The list of formulas is helpful but it is difficult to follow. For example, the parameter "Ihun" is first used in Eq.1 but is only explained in Eq.9. I would strongly recommend starting Chapter 2.2 with a general description of the crop sub-model (preferably with supporting figure) and after that diving into details.

**Response:** Thank you. We restructured the section and described "*Ihun*" upon first mention (Line 119). We also divided section 2.2 into two sub-sections (2.2.1 Overview and 2.2.2 Basic algorithms). Section 2.2.1 provides a general description, and we also referred to the schematic figure of the biophysical process of the model (see response to R2C1). Section 2.2.2 provides the detailed equations.

**R2C7:** Line 165-174: you roughly explain the calibration of "blai" but you do not describe the calibration of "Harvest". As I understand, you first check if changing "blai" is sufficient to reach FAO's yield and only then you check if further changing of "Harvest" is needed, correct? In general, it is not clear how the calibration procedure works.

**Response:** Both *blai* and the harvest index were calibrated at the same time. Similar to the method of Fader et al. (2010), the harvest index was automatically adjusted according to the calibrated *blai*. Respective equations are now listed in Table S1.

**R2C8:** Line 243: Please indicate which models and time period you take from Jägermeyr et al. (2021).

**Response:** Thank you. We added this requested information (Lines 267–269):

*"These countries were selected to make the data comparable with the latest global crop model intercomparison study by Jägermeyr et al. (2021), which includes 11 crop models for the period 1980-2010 (Fig. S10 in Jägermeyr et al., 2021)."*

**R2C9:** Line 248-251: Jägermeyr et al. (2021) were not aiming at representing actual historical crop yields so it is not surprising that their data mostly demonstrates climatic signals.

**Response:** Thank you for the clarification. We understand the study more clearly now.

**R2C10:** Line 254: Double check that you actually take total precipitation and mean temperature for the duration of the growing season and not for the whole calendar year. Also, indicate how you weigh the grid cells here, is it according to harvested areas?

**Response:** We confirmed that we used the total precipitation and mean air temperature for the duration of the growing season. Furthermore, we used the harvest area for weighting.

**R2C11:** Line 276: Instead of referring to a "wide range of regions" you may refer to the percentage of grid cells with significant correlation. Same for areas without correlation.

**Response:** Thank you. We revised the text as follows (Lines 300–305):

*"Using maize as an example (Fig. 8a), statistically significant correlations ($p < 0.1$) were observed in a wide range of regions (e.g., northeastern USA, southern Europe, northeastern China, southern Brazil, eastern Argentina, southern Africa, and eastern Australia) (Fig. 8a), corresponding to 31% of the total grid cells. Notably, there were also substantial differences in a considerable number of locations without statistically significant correlations ($p > 0.1$) (e.g., southeastern USA, western and central Asia, Brazil, and central Africa), corresponding to 69% of the total grid cells (Fig. 8a)."*

**R2C12:** Line 294: There are ways to represent the historical trend in rainfed/irrigated harvested areas. To start, you can scale the total area to FAOSTAT and only then scale the yields. On top of this, you can include historical dynamics in overall rainfed and irrigated croplands as did for maize Mialyk et al. (2022).

**Response:** Thank you for providing this important study. We included it in the discussion (Lines 321–322):

*"To overcome the problems associated with such an assumption, dynamic harvest area data at annual intervals, as generated by Mialyk et al. (2022) should be considered in future studies."*

**R2C13:** Line 137, 343: Please avoid using the word "good" and use academic alternatives instead.

**Response:** We removed the word "good".

**R2C14:** Line 342: Where does the word "bioenergy" come from? You do not use it throughout the text.

**Response:** It refers to the enhancement of bioenergy crop simulation in an earlier study (Ai et al., 2021). However, we agree with the comment and removed text related to "bioenergy".

**R2C15:** Figure 2: It seems that for some countries the improved model shows the same numbers as the default one. Please explain why this is happening in Chapter 3.1.

**Response:** The calibrated model showed nearly the same interannual variation as the default model for some countries because the default simulations were already comparable to yield reported by the FAO, meaning that the calibration resulted in limited improvement. We further clarified this in the text (Lines 261–264):

*"Note that the calibrated model showed a similar performance to that of the default model in some countries (e.g., in USA, France, Ukraine, and Canada for maize) because the default simulations were already comparable to yield reported by the FAO, meaning that the calibration resulted in limited improvement (see Figs. 1a and 2a)."*

**R2C16:** Figure 8: Consider removing it. The message in lines 331-332 is very clear already.

**Response:** Thank you. We removed Fig. 8.

**References:**

Ai, Z., Hanasaki, N., Heck, V. Hasegawa, and T., Fujimori, S.: Simulating second-generation herbaceous bioenergy crop yield using the global hydrological model H08 (v.bio1). Geosci. Model Dev., 13, 6077–6092, https://doi.org/10.5194/gmd-13-6077-2020, 2020.

Ai, Z., Hanasaki, N., Heck, V. Hasegawa, and T., Fujimori, S.: Global bioenergy with carbon capture and storage potential is largely constrained by sustainable irrigation. Nat. Sustain. 4, 884–891 (2021). https://doi.org/10.1038/s41893-021-00740-4, 2021.

Hanasaki, N., Inuzuka, T., Kanae, S., and Oki, T.: An estimation of global virtual water flow and sources of water withdrawal for major crops and livestock products using a global hydrological model. Journal of Hydrology, 384(3-4), 232-244, https://doi:10.1016/j.jhydrol.2009.09.028, 2010.

Heck, V., Gerten, D., Lucht, W. & Popp, A. Biomass-based negative emissions difficult to reconcile with planetary boundaries. Nat. Clim. Change 8, 151–155 (2018).

Dalin, C., Konar, M., Hanasaki, N., Rinaldo, A., Rodriguez-Iturbe, I.: Evolution of the global virtual water trade network. Proc. Natl. Acad. Sci. USA., 109, 16, 5989-5994, https://doi:10.1073/pnas.1203176109, 2012.